# CHALLENGING VLMs' STRUCTURAL SPATIAL INTELLIGENCE THROUGH COMPLEX REASONING TASKS

## ABSTRACT

Large Language Models (LLMs) have undergone rapid progress, largely attributed to reinforcement learning on complex reasoning tasks. In contrast, while spatial intelligence is fundamental for Vision-Language Models (VLMs) in real-world interaction, the systematic study of their complex spatial reasoning remains underexplored. To bridge this gap, we introduce SIRI-Bench, a benchmark designed to evaluate VLMs' structural spatial intelligence through spatial-grounded reasoning tasks. SIRI-Bench comprises 9,000 video-question-answer triplets, where each problem is embedded in a realistic 3D scene. The benchmark is carefully designed so that solving each problem requires both spatial comprehension and structural reasoning. To facilitate large-scale data synthesis, we develop an Automatic Scene Creation Engine that employs collaborative LLM agents to translate abstract mathematical problems into faithful 3D scenes. Experimental results reveal that state-of-the-art VLMs struggle significantly on SIRI-Bench, underscoring the challenge of structural spatial reasoning. We hope that our study will bring researchers' attention to spatially grounded reasoning and advance VLMs in visual problem-solving.

## 1 INTRODUCTION

Recent advancements in Large Language Models (LLMs) and Vision-Language Models (VLMs) have driven the emergence of structural reasoning, enabling strong performance on complex tasks (Wei et al., 2022; Lu et al., 2025; Fan et al., 2025; Jaech et al., 2024; Guo et al., 2025). Through extensive training on mathematics and programming, these models exhibit remarkable generalization across diverse, intricate challenges (Lewkowycz et al., 2022; Achiam et al., 2023; Chen et al., 2024b; Saab et al., 2024; Chen et al., 2024c; 2021b; Li et al., 2022).

With growing interest in complex problem solving, numerous benchmarks have been proposed to assess the reasoning ability of LLMs and VLMs (Hendrycks et al., 2021; Liu et al., 2024; Chen et al., 2024d; Fu et al., 2024). However, most of them focus on abstract tasks such as algebra, program synthesis, or geometric reasoning (Lu et al., 2023; Wang et al., 2024a; Chen et al., 2021a; Zhang et al., 2024a; Amini et al., 2019; Zhang et al., 2024b), largely overlooking visual-based reasoning tasks that are crucial for real-world interaction, particularly, the spatial intelligence.

Spatial intelligence refers to the capacity to interpret spatial information (Gardner, 2011), encompassing skills such as perceiving shapes, understanding transformations, and applying spatial knowledge (Yang et al., 2024c). This capability is essential for the VLMs and downstream real-world deployment (Brohan et al., 2023; O'Neill et al., 2024; Mangalam et al., 2023; Chandrasegaran et al., 2024). While notable studies have explored spatial intelligence (Yang et al., 2024c; Du et al., 2024; Man et al., 2024; Cai et al., 2025), they have largely focused on tasks like distance estimation or layout understanding, which tend to emphasize *intuitive intelligence*. By contrast, relatively little attention has been devoted to the complex reasoning levels that are essential for solving spatially grounded problems, a capacity we refer to as *structured spatial intelligence*.

To address this limitation, we introduce the **S**patial **I**ntelligence **R**eason**I**ng Benchmark (SIRI-Bench), a dataset of 9,078 video–question–answer triplets specifically designed to evaluate VLMs' structural spatial intelligence through complex reasoning tasks. Following the previous paradigm of mathematical structural reasoning (Zhang et al., 2024b; Lu et al., 2023; Wang et al., 2024a; Yue et al., 2024; the University of Utah, 2024; Lightman et al., 2023), we construct SIRI-Bench based on

Figure 1: **Spatial Intelligence with Complex Reasoning.** Conventional benchmarks focus on text-grounded reasoning (upper-left), where reasoning is limited to text alone. In contrast, this work introduces spatial-grounded reasoning (lower-left), where key conditions are implicitly embedded in realistic 3D scenes and presented as videos. Solving these problems requires interleaved textual reasoning and spatial perception (right col.), challenging VLMs' spatial reasoning ability.

solid geometry problems. Unlike conventional text or 2D diagram settings, where conditions are explicitly described (fig. 1 upper-left), SIRI-Bench encodes them in 3D scenes and rendered as videos (fig. 1 lower-left). In this representation, key conditions and spatial relations are implicitly embedded, requiring models to interpret and infer them from spatial cues. This design ensures that both spatial perception and structural reasoning are indispensable for solving the questions, providing a systematic and challenging benchmark for assessing VLMs on spatially grounded reasoning.

Constructing such a dataset at scale entails significant challenges, as manually annotating is highly labor-intensive and demands substantial expertise. To overcome this, we develop an Automatic Scene Creation Engine that translates 3D geometry problems into realistic 3D scenes and renders them into videos. The engine leverages multiple specialized LLM agents within a sophisticated workflow, enabling the automatic computation of object dimensions, Blender scripting, and video rendering. It also adapts problem text to ensure that solving requires genuine spatial reasoning rather than textual shortcuts. As demonstrated by the examples in fig. 3, this engine can process diverse geometry problems and produce faithful 3D scenes, supporting large-scale benchmark creation.

We conduct extensive experiments on SIRI-Bench with a range of popular VLMs. Results show that even SOTA models fail on over 50% of the problems, with prediction errors exceeding 80% compared to ground truth. Remarkably, when key mathematical conditions (e.g. object dimensions and geometric types) are explicitly provided in text, performance improves by more than twofold, indicating models' inability to extract them from video. Further comparisons with human participants and qualitative analyses confirm that current VLMs fall short on SIRI-Bench, underscoring their limitations in spatially grounded reasoning and the value of our benchmark.

The major contributions of this paper are summarized as follows: (1) We introduce the SIRI-Bench, a benchmark designed to investigate VLMs' structural spatial intelligence through complex reasoning tasks. By representing problems through video-based 3D representations, SIRI-Bench establishes a novel framework for evaluating VLMs' structural reasoning capability grounded in spatial comprehension. (2) We develop an Automatic Scene Creation Engine that translates 3D geometry problems into realistic 3D scenes. By leveraging specialized LLM agents with a structured workflow, the engine can produce faithful 3D scenes and significantly reduce the cost of large-scale data generation. (3) We benchmark the performance of state-of-the-art VLMs on SIRI-Bench, and find that they struggle to extract critical spatial information from visual inputs when solving complex reasoning tasks, revealing key limitations in spatially grounded reasoning.

## 2 RELATED WORK

### 2.1 SPATIAL INTELLIGENCE

Spatial intelligence, a core component of cognition, was introduced by Davison (2018) as an extension of visual SLAM (Simultaneous Localization and Mapping). Following this, Chen et al. (2024a) proposed SpatialVLM, trained on 3D visual QA, achieving improved spatial estimation,

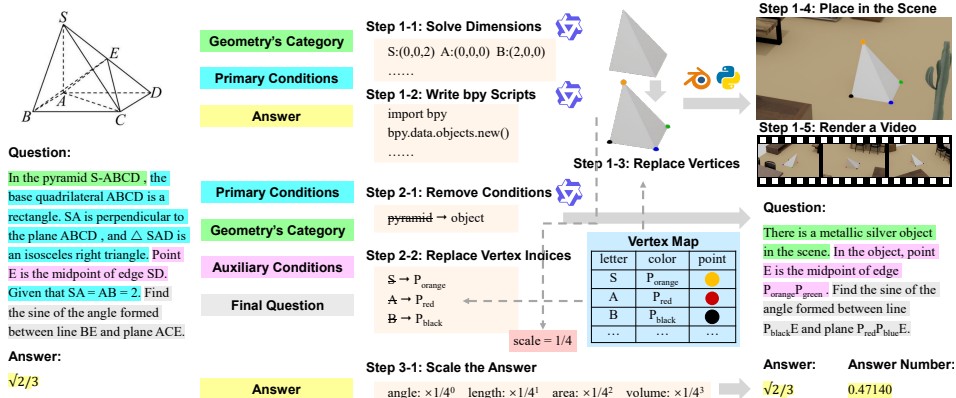

Figure 2: **The Transformation Process** from an original math problem to a 3D Spatial Representation. The given math problem is decomposed into five components and processed individually. First, the main entity's dimensions are solved, and corresponding bpy code generates the 3D scene, later rendered as a video. Second, problem conditions are refined by removing information meant to be inferred from the scene, and node indices are replaced with color markers. Finally, the answer is adjusted for scaling effects to produce the final answer.

while Yang et al. (2024c) defined visual-spatial intelligence as the ability to process visual information in 3D space. Other studies have also introduced benchmarks targeting spatial intelligenceWu et al. (2023b); Yang et al. (2024c); Du et al. (2024); Cai et al. (2025). Despite these advances, current VLMs largely remain at the level of *intuitive intelligence*, such as perception, whereas this study focuses on *structured spatial intelligence* requiring complex reasoning.

## 2.2 COMPLEX REASONING

Complex reasoning involves multi-step inference, abstract thinking, and knowledge integration. Prior work shows that LLMs are capable of such reasoning, mainly through reinforcement learning on mathematics. (Shao et al., 2024; Lu et al., 2025; Guo et al., 2025; Yang et al., 2025). Their applications span mathematics (Lewkowycz et al., 2022; Hendrycks et al., 2021; Achiam et al., 2023; Jaech et al., 2024), medical diagnosis (Chen et al., 2024b; Saab et al., 2024; Chen et al., 2024c), and programing (Chen et al., 2021b; Li et al., 2022; Jaech et al., 2024; Guo et al., 2025), etc. However, LLMs show limited capability in visual-language reasoning (Małkiński et al., 2024; Chia et al., 2024; Ghosal et al., 2024). Although VLMs show proficiency in conventional video question-answering tasks (Wang et al., 2024b; Team, 2024a; Lin et al., 2023; Maaz et al., 2023), their capacity for integrating complex reasoning with spatial intelligence remains underdeveloped. To address this limitation, this paper represents math problems through video-based 3D scenes that jointly demand both spatial understanding and structural reasoning.

## 2.3 VLM REASONING BENCHMARK

A number of benchmarks have been proposed to assess VLMs' visual reasoning, often by reducing linguistic biases and enforcing reliance on visual input (Johnson et al., 2017; Goyal et al., 2017; Hudson & Manning, 2019; Liu et al., 2024; Chen et al., 2024d; Fu et al., 2024). Others, such as MathVista (Lu et al., 2023), MATH-Vision (Wang et al., 2024a) , GeoQA (Chen et al., 2021a), Geoeval (Zhang et al., 2024a), Mathqa (Amini et al., 2019) and MathVerse (Zhang et al., 2024b) , target symbolic and geometric reasoning through plane geometry, but remain abstract and detached from real-world contexts. In contrast, our benchmark is grounded in realistic 3D environments with precise geometric properties (e.g., angles, distances) and requires multi-step inference that integrates spatial perception with procedural reasoning.

## 3 SIRI-BENCH

To evaluate VLMs' spatial intelligence in complex reasoning, we introduce SIRI-Bench, a benchmark built from solid geometry problems and cast as video-based question–answering problems. For

each problem, we elaborately design the 3D spatial representation[*] that embeds key mathematical conditions into realistic 3D scenes, requiring VLMs to extract relevant information through spatial perception and reasoning.

## 3.1 Data Collection

In line with prior work, our benchmark is built upon math problems, as they inherently involve multi-step and structural reasoning (Zhang et al., 2024b; Lu et al., 2023; Wang et al., 2024a; Yue et al., 2024; the University of Utah, 2024; Lightman et al., 2023). The key distinction is that we focus on 3D geometry problems rather than algebra or plane geometry problems, and transform them into realistic 3D scenes for spatial reasoning.

Specifically, SIRI-Bench is constructed from publicly available 3D geometry math problems collected from online educational resources. All problems are translated into English and span difficulty levels from middle to high school. To ensure consistency, we remove low-quality problems, proof-based questions, and other non-numerical items, leaving only problems that require explicit value computation. These are then converted into open-ended problem-solving.

## 3.2 3D Spatial Representation

This section introduces the 3D spatial representation, which encodes geometry problems as realistic 3D scenes rendered into videos. Compared to traditional abstract representations such as textual descriptions or 2D diagrams, our representation offers a closer approximation of real-world scenarios and requires VLMs to extract information from spatial cues rather than textual descriptions.

Specifically, an original geometry problem can be decomposed into five components: (1) *Illustration Diagram*, (2) *Geometry's Category*, (3) *Primary Conditions*, (4) *Auxiliary Conditions*, and (5) *Final Question*, each of which is transformed into the 3D spatial representation through distinct processing steps, thereby depicting the problem anew within the 3D scene.

For the *Illustration Diagram*, it is typically used to visually represent the problem. In our approach, we discard the diagram and displaying the problem through a 3D realistic video.

For the *Geometry's Category*, the main geometric entity is directly presented in the 3D scene. Instead of naming it explicitly (e.g., "the pyramid" or "the cube"), the text only refers to it as "the object" leaving the model to recognize its shape from the visual cue. To avoid information leakage from vertex indices (e.g., "PABCD" indicating a quadrangular pyramid), we replace letter-based indices with color-based indices, with colored points placed in the scene to ensure unambiguous references. For example, the vertex '$S$' in fig. 2 is replaced with the '$P_{orange}$'.

The *primary geometric conditions* refer to the attributes and constraints of the main geometric entity, such as its dimensions and spatial relations. Instead of appearing in the text, these conditions are converted into concrete geometric sizes or spatial positions and directly visualized in the 3D scene, enabling them to be inferred through observation.

The *auxiliary conditions*, which provide necessary but non-central information, are preserved in textual form due to their diverse and complex nature. Similarly, the *final question*, specifying what ultimately needs to be solved, is also kept in textual format.

Our approach is conceptually aligned with MathVerse (Zhang et al., 2024b), which embeds math conditions into 2D diagrams. However, our method goes beyond abstract diagrams by encoding key information in 3D scenes, bringing the problems closer to real-world scenarios. This requires not just optical character recognition (OCR) but genuine spatial perception and problem-solving.

## 3.3 Automatic Scene Creation Engine

Manually converting each problem into a 3D scene is extremely labor-intensive. To facilitate large-scale 3D scene generation, this section introduces an Automatic Scene Creation Engine based on a multi-agent system (Wu et al., 2023a; Hong et al., 2023; Wei et al., 2024; Yang et al., 2024d; Yu et al., 2025). The workflow is shown in fig. 2.

---

[*]The term *3D spatial representation* in this paper refers to representing math problems as 3D scenes. It should not be confused with learnable 3D parameters, such as those used in 3D Gaussian splatting.

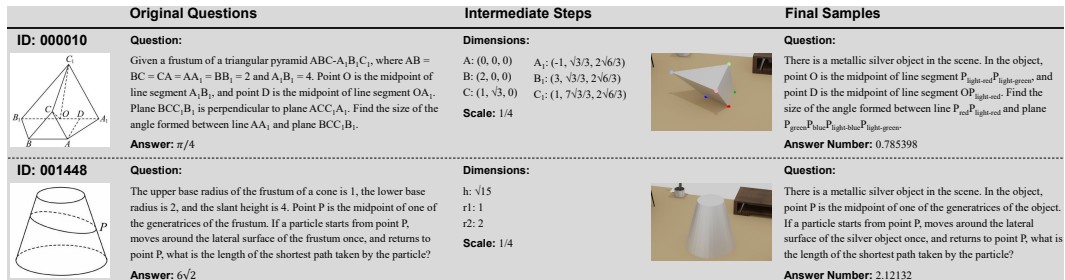

| | Original Questions | Intermediate Steps | | Final Samples |
|---|---|---|---|---|
| **ID: 000010** | **Question:** Given a frustum of a triangular pyramid ABC-$A_1B_1C_1$, where AB = BC = CA = $AA_1$ = $BB_1$ = 2 and $A_1B_1$ = 4. Point O is the midpoint of line segment $A_1B_1$, and point D is the midpoint of line segment $OA_1$. Plane $BCC_1B_1$ is perpendicular to plane $ACC_1A_1$. Find the size of the angle formed between line $AA_1$ and plane $BCC_1B_1$. **Answer:** $\pi/4$ | **Dimensions:** A: (0, 0, 0) B: (2, 0, 0) C: (1, √3, 0) **Scale:** 1/4 | $A_1$: (-1, √3/3, 2√6/3) $B_1$: (3, √3/3, 2√6/3) $C_1$: (1, 7√3/3, 2√6/3) | **Question:** There is a metallic silver object in the scene. In the object, point O is the midpoint of line segment $P_{light-red}P_{light-green}$, and point D is the midpoint of line segment $OP_{light-red}$. Find the size of the angle formed between line $P_{red}P_{light-red}$ and plane $P_{green}P_{blue}P_{light-blue}P_{light-green}$. **Answer Number:** 0.785398 |
| **ID: 001448** | **Question:** The upper base radius of the frustum of a cone is 1, the lower base radius is 2, and the slant height is 4. Point P is the midpoint of one of the generatrices of the frustum. If a particle starts from point P, moves around the lateral surface of the frustum once, and returns to point P, what is the length of the shortest path taken by the particle? **Answer:** 6√2 | **Dimensions:** h: √15 r1: 1 r2: 2 **Scale:** 1/4 | | **Question:** There is a metallic silver object in the scene. In the object, point P is the midpoint of one of the generatrices of the object. If a particle starts from point P, moves around the lateral surface of the silver object once, and returns to point P, what is the length of the shortest path taken by the particle? **Answer Number:** 2.12132 |

Figure 3: **Data Samples in SIRI-Bench.** This figure presents several samples from our SIRI-Bench dataset, along with their original questions and intermediate steps. As can be consistently observed, our data generation engine accurately solves for geometric conditions, replaces vertex indices, processes textual conditions, and computes numerical answers, demonstrating its reliability.

First, the engine begins by parsing the original 3D geometry problem and identifying the main geometric entity. A math-specialist LLM agent (Qwen-Math (Yang et al., 2024b), known for its strong capability on mathematics) is employed to solve for all key conditions necessary to fully define the main geometric entity. After that, all dimensions are scaled to fit appropriately within a unified camera configuration, with sizes ranging from 0.5 to 2m. These parameters are passed to a code-specialist agent (Qwen-LM (Yang et al., 2024a)), which generates bpy scripts(Blender Online Community, 2022) to construct the scene and insert colored vertices (see section 3.2). Second, we refine the problem descriptions to retain only essential information while minimizing textual mathematics. This is achieved by an LLM agent (Qwen-LM (Yang et al., 2024a)), which removes key geometric conditions of the main entity. As discussed in section 3.2, this forces VLMs to rely on spatial comprehension rather than textual cues. Third, the answers are converted from symbolic expressions to numerical values by an LLM agent, allowing for direct measurement of prediction errors and analysis of their distribution. The agent also adjusts values to account for geometric scaling—linear for lengths, quadratic for areas, and cubic for volumes.

By elaborately designing the specialized agents and their workflow, our Automatic Scene Creation Engine is able to transform any 3D geometry problem into a faithful 3D scene. Using this pipeline, we constructed a dataset of 9078 samples for SIRI-Bench, after manually filtering out a few clear conversion errors. While moderate in size, the dataset demonstrates the engine's scalability for generating larger datasets in the future.

### 3.4 OTHER DETAILS

SIRI-Bench consists of 9,078 video–question–answer triplets. Each video is 8 seconds long, recorded at 6 fps with a resolution of 1200×900. The data statistic is shown in Supplementary Materials. For inference, videos are downsampled to 8 or 16 frames and provided as image sequences, a common practice with negligible effect on results Xu et al. (2021; 2024); Marafioti et al. (2025). Videos are rendered by orbiting a camera around the main object along a circular path with an 8 meters radius and a 30 degrees downward tilt. Backgrounds are drawn from 3D-Front (Fu et al., 2021a;b). While the current release adopts a fixed configuration, our data engine supports flexible customization of background, lighting, rendering style, camera setup, and resolution, enabling diverse evaluation settings in future extensions.

### 3.5 DATA SAMPLES IN SIRI-BENCH

In fig. 3, we present illustrative samples from SIRI-Bench, including original questions, final questions and intermediate steps. As shown, our data engine effectively conceals mathematical conditions that must be inferred from video while retaining auxiliary information, and accurately replaces vertex indices with spatially aligned references. Intermediate results further confirm the engine's ability to compute object dimensions, even in complex cases such as sample #000010. Comparisons between original and final answers also validate precise numerical computation with scaling adjustments. Overall, these visualizations demonstrate the reliability of our Automatic Scene Creation Engine in generating faithful 3D scenes from diverse geometry problems.

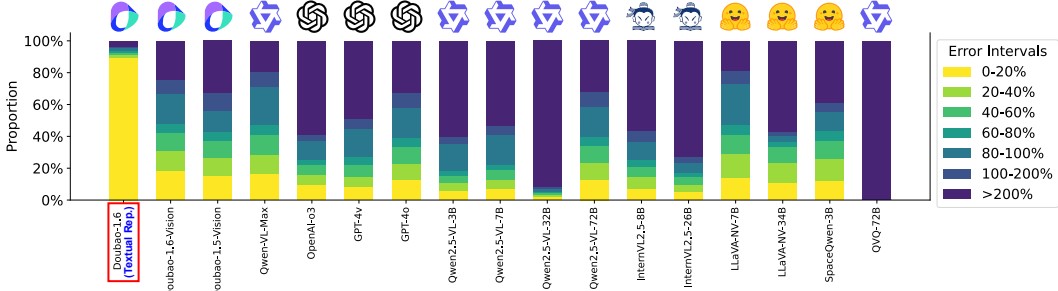

Figure 4: **Performance of Existing VLMs.** This figure shows the error distributions across seven intervals ranging from 0% to 200% for all baseline methods on the SIRI-Bench. A higher concentration of errors in the lower intervals (i.e. brighter colors) indicates better performance in problem-solving. The method labeled 'Textual Rep.' refers to an **LLM that accesses full mathematical conditions** through textual descriptions rather than videos of 3D scenes. Overall, the results reveal the limitations of current VLMs in spatial grounded reasoning.

# 4 EXPERIMENT

## 4.1 EXPERIMENTAL SETUP

**Evaluation Metrics.** Our evaluation aims to jointly assess both spatial estimation accuracy and mathematical reasoning correctness. To achieve this, we compute the relative error between the predicted numerical answer and the ground-truth value, which can be expressed as: $\frac{|\hat{y}-y|}{y} \times 100\%$, where $\hat{y}$ denotes the predicted numerical value and $y$ denotes the ground-truth numerical value. To better analyze performance, we categorize the relative error into seven intervals (0–20%, 20–40%, 40–80%, 80–100%, 100–200%, and >200%) and examine the distribution of predictions. Strong models concentrate in lower-error intervals, whereas weaker models show higher errors.

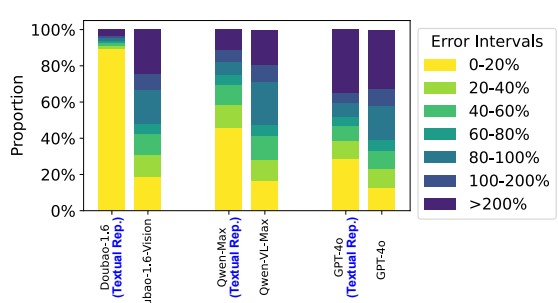

Figure 5: **Ablation on Problem Representation.** This figure compares the accuracy of two sibling models using textual representation versus 3D spatial representation as input. Three columns depict three pairs of sibling LLMs/VLMs. This comparison disentangles structural reasoning from spatial perception, revealing that existing VLMs struggle to effectively extract spatial information when solving complex visual problems.

**Baselines.** We evaluated a variety of VLMs, encompassing models of different sizes, both open/close-source models. They are[*]: **OpenAI:** OpenAI-o3, GPT-4o, GPT-4v OpenAI (2023b; 2024; 2023a). **Qwen:** Qwen2.5-VL-7B, 32B, 72B, QVQ-72B, and Qwen-VL-Max (Yang et al., 2024a; Team, 2024b;c). **Doubao:** Doubao-1.6-vision, Doubao-1.5-vision (Team, 2025b;a). **InternVL:** InternVL2.5-8B, 26B (Chen et al., 2024e). **LLaVA-Next-Video:** LLaVA-NV-7B, 34B (Zhang et al., 2024c). **Others:** SpaceQwen-3B (Chen et al., 2024a). **LLM:** Doubao-1.6 (Textual Rep.) (Team, 2025b), which accesses the full mathematical conditions through text rather than videos. We append it to disentangle structural reasoning from spatial perception.

## 4.2 OVERALL PERFORMANCE

The overall performance of existing VLMs is summarized in fig. 4, which presents the distribution of prediction errors across seven intervals ranging from 0% to over 200%. A higher concentration of predictions in the lower error intervals indicates greater accuracy.

All evaluated VLMs perform poorly on SIRI-Bench. The best method, Doubao-1.6-Vision (Team, 2025b), achieves only 31% of predictions within a 0–40% error margin. This performance is consistent with its strong results on other multimodal reasoning benchmarks (Wang et al., 2024a; Lu et al., 2023). Nevertheless, over 33% of its predictions exceed 100% error, underscoring its sub-

---

[*]The specific versions we used is o3-2025-04-16, gpt-4o-2024-08-06, gpt-4-turbo-2024-04-09, qwen-vl-max-2025-08-13, qwen-max-2024-09-19, doubao-seed-1-6-vision-250815, doubao-1.5-vision-pro-250328

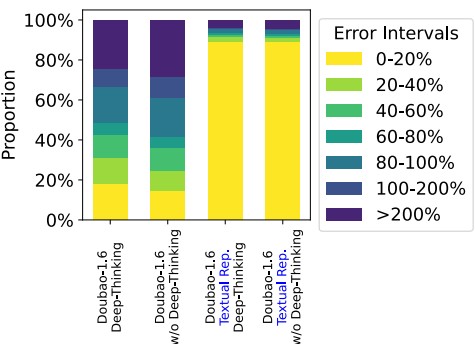

Figure 6: **Ablation on Deep-Thinking.** This figure compares the performance with or without Deep-Thinking. The left two columns use 3D spatial representation and the right two use textual representation.

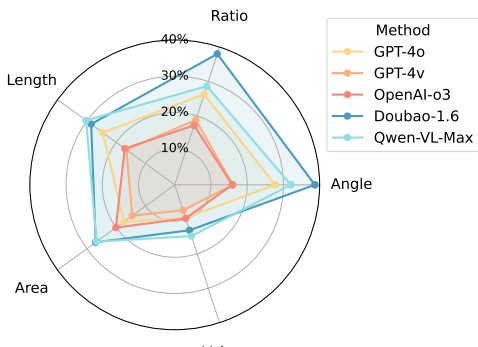

Figure 7: **Error Distribution by Problem Type.** The figure shows the success rate of five baseline models across five types of problems, where success is define relative error below 40%.

stantial limitations. (Doubao-1.6-Textual-Rep. is discussed separately in section 4.3.) There is no evident correlation between performance and parameter size. For instance, Qwen2.5-VL improves from 3B to 7B, but InternVL2.5 declines when scaling from 7B to 26B. This implies that enhancing the spatial grounded reasoning ability may need other strategies beyond simply scaling up. Another noteworthy case is SpaceQwen-3B (Chen et al., 2024a), a small model fine-tuned for spatial perception. It achieves 37% of predictions under 60% error, showing competitive performance relative to larger models. However, its accuracy remains unsatisfactory, likely due to both its limited scale and a fine-tuning focus on intuitive perception rather than structural reasoning.

We hypothesize two main reasons for their poor performance: (1) models struggle to estimate object dimensions and spatial relationships since they are not explicitly provided; (2) models are prone to errors in mathematical reasoning and calculation. We will further analyze these issues in later sections. In summary, the consistent failure across models underscores the limitations of current VLMs in structural spatial reasoning.

### 4.3 ABLATION ON INPUT REPRESENTATION

As shown in fig. 4, the doubao-1.6-Textual-Rep. substantially outperforms all VLM baselines. This method is an LLM model that inputs textual representations and accesses mathematical conditions through text. To further investigate this phenomenon, we compare three pairs of sibling models: one LLM using text input (i.e. textual representation) and one VLM using video-text input (i.e. 3D spatial representation). The only difference lies in whether spatial information is directly presented in text or must be derived from video. This ablation disentangles structural reasoning from spatial perception, isolating the source of performance gaps.

The results in fig. 5 demonstrate a consistent trend: text-input models clearly outperform their video-input counterparts. For example, the proportion of predictions within 20% error nearly doubled when switching from video to text input (18.4%, 16.4%, 12.7% vs. 89.2%, 64.0%, 28.9%). These findings indicate that while SOTA LLMs can handle complex reasoning when conditions are provided in language, VLMs fail to do the same on the visual level, as they struggle to extract sufficient spatial information from video. Moreover, the strong performance of Doubao-1.6 validates the effectiveness of SIRI-Bench, as most questions are solvable with full textual conditions.

### 4.4 ABLATION ON DEEP-THINKING

We conduct an ablation study to examine the effect of Deep-Thinking, which refers to an explicit deep reasoning phase before producing the final answer. As shown in fig. 6, when the input is 3D spatial representation, disabling Deep-Thinking for Doubao-1.6 reduces performance from from 31% to 25% (prediction error below 40%). In contrast, the impact is minimal with textual input, likely because direct answering is sufficient for most problems. These results suggest that deep reasoning plays a more critical role in spatial grounded reasoning.

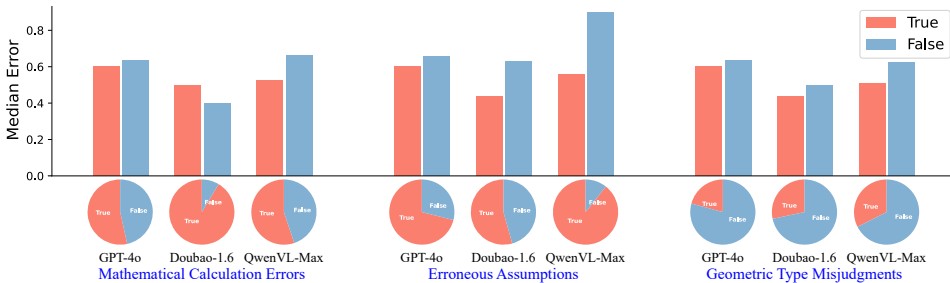

Figure 8: **Analysis of Reasoning Mistakes and Their Impact.** The lower part shows the proportion of VLM reasoning responses exhibiting each of the three mistakes, where **True** means no mistake and **False** otherwise. The upper part illustrates the median prediction error associated with each condition, revealing how specific reasoning mistakes affect final performance.

## 4.5 ERROR DISTRIBUTION

We analyze model performance across 5 problem types: angle, ratio, length, area, volume. We define a prediction as successful if its relative error is below 40% and report 5 corresponding success rates in fig. 7. There are two major observations. (1) All methods perform poorly on volume-related problems, with fewer than 20% of their predictions under the 40% error threshold. We suggest that errors in volume problems may stem from spatial estimation, as the deviations will be amplified cubically in the calculation. (2) Among the five baselines, Doubao-1.6-Vision achieves the best overall performance. Its advantage is mainly on angle and ratio problems, where the performance improves from 30% to 40% over the second-best model. Since solving angle and ratio questions primarily requires reasoning about spatial relations rather than absolute dimensions, this finding highlights Doubao-1.6's strength in relational geometric reasoning and suggests room for improvement for other models in this aspect.

## 4.6 FULL RESPONSE ANALYSIS

We further analyze VLMs' full reasoning response by examining three types of mistake, mathematical calculation errors, erroneous assumptions, and geometric type misjudgments. Specifically, we provide VLMs' full reasoning response to another LLM, which identifies whether any of these mistake occur. For geometric type mistake, the correct geometric type is also provided. We then visualize and study how these mistakes correlates with the error distribution in fig. 8.

Across all three categories, the occurrence of any mistake consistently leads to higher prediction errors. The most notable case is erroneous assumptions, where the median error increases from 0.56 to 0.90 for Qwen-VL-Max. Moreover, VLMs often misidentify the geometric type. For instance, GPT-4o exhibits such mistakes in nearly 79% of the problems. These results indicate that failures in geometric type recognition and the tendency to introduce unsupported assumptions are key factors limiting VLMs' reasoning accuracy.

## 4.7 COMPARISON WITH HUMAN PERFORMANCE

In this part, we investigate how far VLMs are from matching human performance. We randomly select seven samples from SIRI-Bench and ask participants to solve them without additional information. The average error distribution of human participants and VLMs is shown in fig. 9. As can be seen, humans participants successfully solve over 30% problems with errors within 20%. In contrast, none of the VLMs achieve this level of accuracy. When allowing for a 60% error margin, humans succeed on about 70% of the problems, far surpassing VLMs. Yet, some problems remain challenging even for humans, occasionally yielding errors above 100%, likely due to accumulated deviations in distance estimation. Nevertheless, the overall results demonstrate that current VLMs consistently fall short of human capability.

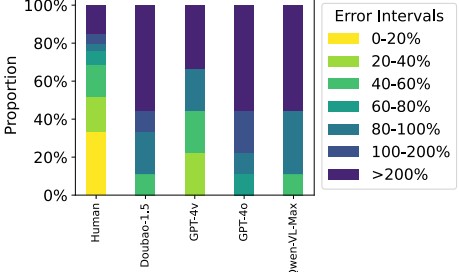

Figure 9: **Comparison with Human Performance.** This figure compares the performance of human participants (the 1st bar) with four VLMs on a subset of the SIRI-Bench, showing that current VLMs are still far from matching human performance.

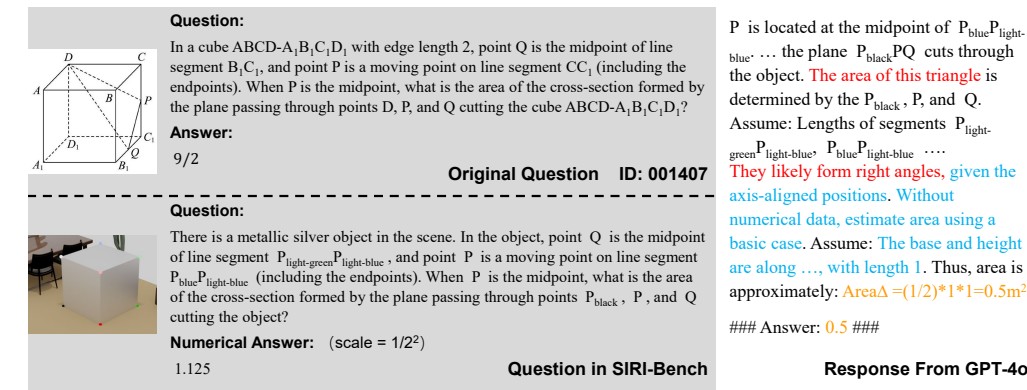

Figure 10: **Display of GPT4o's Full Answer.** This figure displays the full response from GPT-4o on an example, with spatial interpretation errors, spatial perception errors, and the final incorrect result highlighted in red, blue, and orange, respectively. The visualization illustrates that GPT-4o struggles to extract problem-solving conditions from visual inputs, and tends to make unjustified assumptions that ultimately lead to incorrect answers.

## 4.8 CASE STUDY

Figure 10 presents a case study of GPT-4o's reasoning process, with key errors highlighted. This case illustrates potential factors contributing to GPT-4o's poor performance on SIRI-Bench. Specifically, the question asks for the area of the cross-section formed by the plane defined by points D, P, and Q intersecting a cube. GPT-4o oversimplifies the problem by assuming that three points define a triangular cross-section (DPQ), whereas the actual cross-section is a rectangle. Additionally, without effectively extracting spatial cues from the video, it further assumes the triangle is right-angled and estimates the cube's size as unit-length, ultimately leading to an incorrect answer.

Overall, this example highlights GPT-4o's difficulty in interpreting geometric constraints and its tendency to make unjustified assumptions when visual cues are implicit. Rather than leveraging the video to resolve ambiguity, GPT-4o often proceeds with incomplete reasoning, revealing a notable problem-solving limitation in the visual domain.

## 5 CONCLUSION

This paper presents SIRI-Bench, a benchmark designed to assess the structural spatial intelligence of VLMs through 3D mathematical reasoning problems. SIRI-Bench consists of 9078 video-question-answer triplets, where each question is formulated within a realistic 3D scene captured in video format. Solving these questions requires models to extract, interpret, and perform reasoning based on spatial cues, offering a novel and rigorous evaluation of VLMs' spatial grounded reasoning ability. To construct this dataset efficiently, we develop an Automatic Scene Creation Engine that converts abstract math problems into realistic 3D scenes and renders corresponding videos. The engine coordinates multiple specialized LLM agents within a structured workflow, ensuring reliable scene generation while significantly reducing the cost of large-scale data production. Experiments on SIRI-Bench demonstrate that existing VLMs consistently fail to solve more than 50% problems when spatial information must be inferred from video. Their performance not only falls far behind human level but also lags significantly behind their text-only counterparts, which are provided with explicit mathematical conditions in text form. Further experiments reveal that existing VLMs over-rely on textual cues and struggle to actively and accurately extract key spatial information—such as object types, geometric dimensions, and spatial relations—from visual input. These findings highlight a fundamental limitation of current VLMs and demonstrate the value of SIRI-Bench[*].

---

[*]Upon acceptance, we will release the code and dataset for public access

## ETHICS STATEMENT

Our work is conducted in accordance with ethical research practices and legal requirements. All original mathematical problems used in SIRI-Bench are licensed for research use, and the 3D scenes are generated based on the publicly available 3D-FRONT dataset, for which we have obtained proper authorization. The data generation process involves only synthetic scenes and automatically rendered videos, without collecting or inferring any personally identifiable information (PII). We emphasize that SIRI-Bench is designed solely for academic research on spatial reasoning and model evaluation. It should not be misused for surveillance, discriminatory applications, or other purposes that could harm individuals or communities.

## REPRODUCIBILITY STATEMENT

We provide comprehensive details of our dataset construction pipeline. All prompts used in data generation and experiments are included in the Appendix to enable precise replication. We also describe our experimental settings, evaluation protocols, and ablation study configurations in detail. We will publicly release the SIRI-Bench dataset, along with the associated code and generation scripts, upon acceptance. This ensures that all results presented in the paper can be independently verified and extended by the research community.

## LLM CLARIFICATION

All uses of Large Language Models (LLMs) in this work are explicitly documented in the paper. During dataset construction, they were used solely as auxiliary tools for repetitive tasks such as condition text editing and dimension solving, always under human supervision. Crucially, all core scientific contributions—including the formulation of the research problem, the design of the SIRI-Bench dataset, the development of the scene generation pipeline, and the experimental design and analysis—were conceived and executed by the authors. The use of LLMs did not involve conceptualization, methodology design, or result interpretation, and did not reach a level that would warrant considering them as contributors.

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

# Challenging VLMs' Structural Spatial Intelligence through Complex Reasoning Tasks

# Supplementary Materials

## A. DATA RELIABILITY

Ensuring the reliability of data is essential for a benchmark designed to evaluate spatial reasoning. We validate the reliability of SIRI-Bench from three key perspectives: geometric dimension solving, condition text processing, and numerical answer correctness.

### Geometric Dimension Solving

To assess the reliability of geometric dimensions solved by our Automatic Scene Creation Engine, we sample 70 cases (specifically, first 70 cases) from the dataset and asked human participants to evaluate the results. Each participant watch the rendered video of a scene and compared it against the original schematic diagram of the problem. A case is marked as correct if the rendered geometry matched the schematic and incorrect otherwise. The average correct rate across all cases is $81.4\% \pm 1.4\%$. Yet, the actual reliability is likely higher, since the schematic serves only as an illustrative reference and is often more restrictive than the original problem specification.

### Condition Text Processing

To remove key spatial information from problem conditions while preserving auxiliary details, we employ Qwen-LM [1] with a carefully designed prompt (see section F). The model is strictly instructed to remove only the information related to the primary geometric object, ensuring that essential supporting information remains intact. This design means that even if Qwen-LM occasionally leaves some targeted information undeleted, it will never mistakenly remove essential information, thus preserving the solvability of every problem.

Additionally, we replace letter-based vertex indices with color-based indices using a deterministic regular expression–based algorithm. The deterministic nature of this replacement process ensures the consistency and reliability of indices replacing.

### Numerical Answer Correctness

Validating the correctness of mathematical answers is both critical and challenging. Existing tools such as math_verify [2] can determine whether two symbolic expressions are mathematically equivalent. However, we choose to convert symbolic expressions into numerical values using Qwen-LM [1]. This decision is motivated by two considerations: (1) Previous work (including math_verify) reports comparable reliability to LLM-based approaches like Qwen-LM [2]. (2) Compared with true-or-false judgement, numerical outputs enable direct computation of relative errors, which is crucial for evaluating models' spatial perception accuracy in our experiments.

## B. DATA DISTRIBUTION

Figure S1 illustrates the distribution of problems within our dataset across different geometric types and problem types. The geometric entity types refer to the primary geometric entity involved in the problems, with a total of 13 distinct types. Problem types denote the specific quantities that the problems require solving for, including angle, ratio, length, area, volume. As depicted in fig. S1, our dataset achieves a balanced distribution across all geometric entity types and problem types. This balanced distribution ensures that our dataset can comprehensively evaluate the performance of VLMs on a wide range of problems, providing a robust benchmark for assessing their capabilities in various contexts.

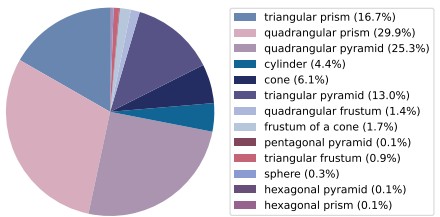
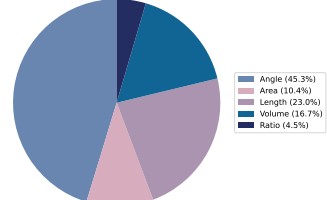

**Fig. S1. Data Distribution.** The SIRI-Bench dataset ensures a balanced distribution across geometric types (left), and problem types (right).

## C. MORE DATA VISUALIZATIONS

We provide 70 additional data samples in the supplementary file, together with their corresponding videos and intermediate results. As can be consistently observed across these samples, our data generation engine accurately solves for geometric conditions, replaces vertex indices, processes textual conditions, and computes numerical answers, demonstrating its robustness and reliability.

## D. WHY QVQ PERFORMS SO POORLY

As illustrated in Figure 1 of our main paper, QVQ [3] demonstrates the lowest performance among all evaluated visual-language models (VLMs) on the SIRI-Bench benchmark. Given QVQ's design as a multimodal reasoning model with advanced visual understanding capabilities, this underperformance warrants further investigation.

QVQ [3] is built upon the Qwen2-VL-72B[4] architecture and is tailored for complex visual reasoning tasks, including mathematical problem-solving. It has achieved commendable results on benchmarks such as MMMU, MathVista, and MathVision, showcasing its potential in handling intricate visual and mathematical challenges. Notably, QVQ is designed to emulate human-like reasoning by engaging in step-by-step analysis and self-reflection during problem-solving. However, when applied to SIRI-Bench, which presents novel 3D geometry problems requiring spatial reasoning, QVQ struggles significantly.

Upon examining its reasoning processes, we observe that QVQ often engages in extended chains of thought, **spanning up to 8K or even 16K tokens, without arriving at a definitive conclusion.** This behavior suggests that QVQ becomes entangled in recursive reasoning loops, failing to produce final answers. Consequently, its responses are categorized as having errors exceeding 200%, contributing to its poor performance metrics.

## E. LIMITATION AND FUTURE WORK

Although our benchmark has uncovered several insightful findings under the current experimental settings, all videos are currently generated under a fixed camera configuration and consistent environment conditions. This design ensures a unified comparison but limits the diversity of visual input.

Given the flexibility of our data generation engine, it can easily create videos with varied object scales, backgrounds, and lighting conditions for the same underlying problem. Future work could introduce more diverse realistic 3D scenes to enable broader evaluations of VLMs It could also explore interactive settings where VLMs navigate 3D environments to acquire spatial information, which would be highly intriguing.

## F. FULL PROMPTS

This section presents all the detailed prompts used in our study.

### Inference Prompt

During VLM's inference, we use the following prompt to guide model outputs:

```
Solve the following solid geometry problem.  Think step by step, estimate and
find the numerical answer.  Use meters as the basic unit for length, and use
radians for angles.  In the last line of your answer, provide the final answer.
The final answer should be a numerical value and must not contain formulas,
symbols, unknown quantities, or references to unknown quantities.  The last line
of your response should be of the form Answer:  $ANSWER (without quotes) where
$ANSWER is the answer to the problem with no
boxed command.
{question}
Remember to put your answer on its own line after "Answer:", and do not use
boxed command.
```

### Neural Translation Prompt

We use the following prompt to translate the mathematical problem from Chinese to English:

```
I will provide you with a math problem described in Chinese, usually a geometry
problem.  Your task is to translate this problem into English.  You do not
need to solve this math problem.  You only need to translate it into English.
Respond with the English translation directly, with no additional content.
Now, here is the Chinese geometry problem:
{question_zh}
Please translate it.
```

### Dimension-Solving Prompt

In our Automatic Scene Creation Engine, we employ a math-specialist agent (Qwen-Math [5]) to solve for all key conditions necessary to fully define the main geometric entity.

```
I will provide you with a 3D geometric solid along with its conditions.  Your
task is to calculate the specific dimensions of this geometric solid.  If the
given conditions do not uniquely determine the solid, you may output any one
valid solution that satisfies the constraints.  Do not include units in your
answer.
Here, the given 3D geometric solid is:  {geometry_type}.
The given conditions of this geometric solid are:
{condition}.
Your task is to {solving_target}
After completing the reasoning and calculations, summarize the final result in
the following format:
Final Answer:  xxx
In your final answer, when giving values, please clearly label what they
represent.  For example:
Final Answer:  'The coordinate of [point name] is [value]', 'The value of
[dimension] is [value]'.
```

### Condition-Processing Prompt

In our Automatic Scene Creation Engine, we refine the problem descriptions to retain only essential information while minimizing textual mathematics. This is achieved by an LLM agent (Qwen-LM [1]), using the following prompt.

```
Please do the step by step reasoning.
You will be given a description that is part of a 3D geometry problem.  In this
description, there is a {geometry_type} that serves as the main geometric object
(it may not always be referred to as '{geometry_type}' but could appear under
other related names).  And the description also contains other entities such as
points or line segments.
Your task is to carefully determine which parts of the description describe
the main geometric object and which describe other entities, then remove all
information about the main geometric object while strictly preserving all
information about the other entities.
Specifically, the main geometric object (i.e.  the {geometry_type}) is defined
by {n_vertex} vertices.  Any points, lines, or planes composed exclusively of
these {n_vertex} vertices belong to the main geometric object, and statements
concerning only these elements must be removed, including their dimensions,
shapes, geometric characteristics, etc.  Any entities composed entirely of other
elements, or partly of these {n_vertex} vertices together with other elements,
are considered outside the main geometric object, and statements about them
must be kept.  If a statement involves both the main geometric object and other
entities, keep it, but replace any names or references to the main geometric
object with 'the object,' ensuring the statement contains no words that reveal
its shape (e.g., polyhedron, cube, prism, pyramid, frustum, etc.).
Please do the step by step reasoning; first correctly identify the main
geometric object and the vertices that define it; then determine whether each
entity belongs to the main geometric object or not; then split the passage
into as fine-grained sub-statements as possible and carefully evaluate each
sub-statement; then perform the removals and retentions; and finally summarize
the processed result in the following format:
Final Answer:  the description text with some information removed.  (Do not
include any explanations or enumeration symbols in the final answer, but rather
provide a coherent descriptive sentence.)
Now, here is the given description:
{condition}.
Please process it.
```

*Solving Numerical Answer Prompt*

In order to measure prediction errors and analyze their distribution, we convert the answer from
symbolic expressions to Python expressions by Qwen-LM [1], using the following prompt. The
Python expressions are then converted to numerical value.

```
You will be given a mathematical expression that may include formulas or symbols
written in Markdown, LaTeX, or other formats.  Your task is to accurately parse
and convert the expression into a single valid Python expression using only the
standard math library.  Please do step-by-step reasoning and accurately carry
out the conversion.
The following specific requirements apply:
- Do not include any units in the final expression.
- Treat angles as radians by default; if an angle is explicitly given in degrees,
wrap it with math.radians(...).
- Do not use libraries other than math.
- After your reasoning, provide the final answer in this format, containing only
the Python expression with no extra words, symbols, or units:
Final Answer:  python_expression.
Now here is the given expression:
{answer}.
Please convert it.
```

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
