# OpenReview forum: "Challenging VLMs' Structural Spatial Intelligence through Complex Reasoning Tasks"
_ICLR.cc/2026/Conference — ICLR 2026 Conference Withdrawn Submission_

### Official Review · Reviewer_Ld4Z · 2025-10-30

**Soundness:** 2
**Presentation:** 3
**Contribution:** 2
**Rating:** 4
**Confidence:** 4

**Summary:**

This paper introduces a spatial reasoning benchmark SIRI-Bench constructed from existing 3D math problems and used a 3D data engine for video generation. The paper evaluated several VLMs including both open-weight and API models, the results demonstrated the challenging nature of the benchmark as well as the requirements for visual understanding beyond just textual comprehension.

**Strengths:**

1. The benchmarks is about 3D reasoning is in-time and poses a relatively (for open-weight models) challenging set for test
2. The evaluation is comprehensive, including some ablations demonstrating aspects of model error and the need of visual perception beyond just language reasoning

**Weaknesses:**

1. The data source is limited, as all questions are from 3D math problems (Sec 3.1). But there are other scenarios for 3D reasoning in real-world applications, such as navigation, physics problems
2. All data are synthetic in just one style (since you are using one configuration of one data creation engine). A real and applicable benchmark should at least involve some of the real-world visual data
3. Missing some of the most powerful VLMs such as o series and GPT-5 from OpenAI and Google’s Gemini family. Most of current evaluated models are open-weight, thus the claim of ‘challenging’ maybe jeopardized due to the limited powerful models

**Questions:**

1. In sec 4.4, you only tested Doubao-1.6 with different thinking efforts, will the claim still hold if using other models with different thinking budgets? E.g., GPT-5 from OpenAI or Claude from Anthropic
2. Since the data is fully synthetic, have you manually checked these data after its production to make sure the label and annotations are valid?
3. Sec 4.6, what is the LLM for judging the error type? Why choose this one? Do you have a correlation between this judgement with human perceptions?

---

### Official Review · Reviewer_BGxw · 2025-10-30

**Soundness:** 3
**Presentation:** 3
**Contribution:** 2
**Rating:** 4
**Confidence:** 4

**Summary:**

This paper introduces SIRI-Bench, a benchmark designed to evaluate the structural spatial intelligence of Vision-Language Models (VLMs).
The authors build a dataset of 9,078 video–question–answer triplets derived from 3D geometry problems, generated automatically via an Automatic Scene Creation Engine that converts symbolic math problems into realistic 3D scenes.
Extensive experiments across more than ten VLMs (GPT-4o, Qwen-VL, Doubao-Vision, etc.) show that current models perform poorly on this benchmark, highlighting a significant gap between textual reasoning and spatially grounded reasoning.

**Strengths:**

1. The paper thoroughly evaluates a wide range of state-of-the-art VLMs under various setups (textual vs. spatial input, deep-thinking ablations, human comparison, etc.), producing a valuable analysis of their spatial reasoning capabilities.
2. The Automatic Scene Creation Engine is well-engineered, and the dataset design (colored vertex markers, removal of text-based geometric clues) reflects careful thought.
The released benchmark would likely become a useful resource for the community.
3. The paper is well-written, and sufficient visualization and analysis make the paper easy to follow.

**Weaknesses:**

1. The paper provides a useful benchmark but lacks methodological novelty.
It mainly offers engineering and analysis contributions without proposing improvements or insights for advancing VLM spatial reasoning.
2. The benchmark targets structured geometric problems with explicit parameters.
There is little discussion on extending to unstructured or open-world spatial reasoning tasks.
3. The dataset uses fixed camera and lighting conditions with static 3D-Front backgrounds.
Although the engine can vary rendering factors, no experiments or analysis explore their impact on robustness or generalization.

**Questions:**

I appreciate the authors’ dataset contribution and the thorough experimental analysis.
My main concern is that the paper stops at empirical observation without providing insights or guidance for future improvements.
It would be stronger if the authors could extract conceptual takeaways or design implications from the results.

As for the dataset, since it relies on rendering, further discussion on how irrelevant variables (such as background and lighting) impact model performance would be valuable. Additionally, exploring how the structured approach could be generalized beyond geometric problems to more realistic, unstructured environments would contribute significantly to the community.

---

### Official Review · Reviewer_ATTz · 2025-11-01

**Soundness:** 2
**Presentation:** 3
**Contribution:** 2
**Rating:** 4
**Confidence:** 4

**Summary:**

This paper converts geometric math problems into realistic 3D scenes using Blender. Experiments on established benchmarks show that state-of-the-art vision-language models fail on more than 50% of these problems. Interestingly, when certain key mathematical conditions are explicitly revealed in the text, the models’ performance improves by more than twofold.

**Strengths:**

- The idea of bridging geometric mathematical reasoning problems to real-world 3D scenes is novel and valuable.
- The paper provides a comprehensive full-response analysis (Section 4.6), which adds depth to the evaluation.

**Weaknesses:**

- Additional qualitative and response-level analyses would be helpful (see questions below), especially to better understand how and why the models fail.

**Questions:**

- Why are the problems rendered as videos rather than single images or multi-view images?
- Since the major challenge lies in perceiving 3D geometry, could the authors include direct comparisons between reasoning and non-reasoning models?
- The ablation study in Section 4.3 is insightful, but is it possible to control the amount of key mathematical information revealed? This could help identify which specific pieces of information are misinterpreted by VLMs.
- In Figure 4, moving from Qwen2.5-VL 3B to 32B leads to a performance drop. Could the authors provide more interpretation or examples of how Qwen2.5-VL 32B behaves differently? Conversely, why does the 72B variant recover strong performance?
- Figure 9 includes human performance results, which is useful but insufficient to gauge how good or bad the current SoTA VLMs are. Could the authors add a simple baseline, such as a model that predicts a constant answer or other heuristic baselines, for better comparison?

---

### Official Review · Reviewer_eMUH · 2025-11-05

**Soundness:** 3
**Presentation:** 3
**Contribution:** 2
**Rating:** 4
**Confidence:** 3

**Summary:**

This paper proposes SIRI-Bench, a benchmark designed to evaluate the multi-step structural spatial reasoning ability of VLMs under video-based 3D representations. The benchmark is automatically generated by an Automatic Scene Creation Engine that translates 3D geometry problems into realistic 3D scenes using a structured multi-agent LLM workflow. Experimental results show that current VLMs fail to effectively extract geometric conditions from visual input, even when their underlying LLM components demonstrate strong reasoning capabilities on textual equivalents.

**Strengths:**

- The paper introduces a technically sound automatic data generation framework based on a multi-agent pipeline, avoiding manual annotation.

- The inclusion of human baselines and visualized reasoning paths in Figure 10 adds interpretability, and the results demonstrate that VLMs fall short of human capability.

- The experiments are comprehensive, covering a wide range of VLMs and LLMs.

**Weaknesses:**

- The task setup is somewhat vague. In many problems, the model is expected to estimate geometric quantities such as edge lengths directly from the rendered scene, but the visual input lacks clear reference objects information (e.g., chair or table height in the image) that would allow for reliable inference. This ambiguity may inherently limit model performance, regardless of reasoning ability.

- The finding that spatial reasoning bottlenecks mainly stem from limited visual perception capability is already a well-known observation. The paper does not uncover substantially new insights into why or how this limitation arises.

- The paper focuses primarily on diagnosing this phenomenon through the benchmark, but lacks actionable insights or attempts to mitigate or address the problem, for example, through prompt design, training, or architectural modification.

**Questions:**

- Regarding Weakness 1, could the authors test whether providing explicit reference objects' information (e.g.,  chair or table height) improves model performance?

- It would be interesting if the authors could also report the same problem type-wise error breakdown shown in Figure 7 for the textual representation LLMs.

- In Section 4.6, how did the authors ensure that the three types of mistakes (mathematical, geometric, and assumption errors) identified by the LLM were correctly classified?

---

### Note · Authors · 2026-01-29

**Comment:**

We greatly appreciate the valuable comments and suggestions provided by the reviewers on this paper.

**Withdrawal Confirmation:**

I have read and agree with the venue's withdrawal policy on behalf of myself and my co-authors.

---

### Meta-Review · Area_Chair_rkyF · 2026-01-02

**Summary:**

The key reviewer concerns are:

* Limited methodological novelty: primarily a benchmark without actionable techniques to improve VLM spatial reasoning

* Potential task ambiguity: lack of explicit reference scales may cap performance (independent of reasoning ability)

* Restricted scope and realism: fully synthetic, single style, fixed rendering choices; limited discussion of robustness/generalisation; narrow source domain (3D math problems)

* Missing or incomplete comparisons: no coverage of strongest closed models; limited ablations; unclear validation of error-type judgments

As outlined below, no author response was visible, leaving these concerns unresolved. With a consistent negative lean from the reviewers, I do not think there is cause for overruling this, so I propose reject.

**Reviewer Concerns:**

The contribution is primarily an engineering / benchmarking effort: the dataset and pipeline are careful, but the work does not extract design insights or propose methods to improve spatially grounded reasoning, so its significance hinges on diagnostic clarity.

Several reviewer issues remain unaddressed: (i) task ambiguity from missing reference cues, making it hard to separate perception limits from ill-posedness; (ii) limited realism and robustness analyses due to a fully synthetic setup; (iii) incomplete comparisons and validations, including reasoning vs. non-reasoning modes, controlled information ablations and missing error taxonomies; and (iv) lack of empirical justification for video over single or multi-view images. In the absence of a visible rebuttal, these gaps persist.

Overall, the work offers a potentially useful benchmark and thorough baseline evaluation across many open-weight models. However, the unresolved issues above limit the paper’s current significance: the benchmark’s diagnostic value is undercut by potential ambiguities, and the lack of broader coverage and robustness analyses reduces confidence in generality.

**Reviewer Scores:**

All four reviewers are at 4 (marginally below acceptance), with moderate confidence. No rebuttal was provided, there was no clear movement toward a positive consensus, and key answers and additions were not supplied during discussion. I therefore recommend rejection at this time, while encouraging the authors to strengthen the work for a future submission.

---

### Decision · Program_Chairs · 2026-01-26

Reject